# High-resolution crossover mapping reveals similarities and differences of male and female recombination in maize

Penny M.A. Kianian[1], Minghui Wang[2,3], Kristin Simons[4], Farhad Ghavami[4,7], Yan He[2,8], Stefanie Dukowic-Schulze [1], Anitha Sundararajan[5], Qi Sun[3], Jaroslaw Pillardy[3], Joann Mudge[5], Changbin Chen[1], Shahryar F. Kianian[6] & Wojciech P. Pawlowski[2]

Meiotic crossovers (COs) are not uniformly distributed across the genome. Factors affecting this phenomenon are not well understood. Although many species exhibit large differences in CO numbers between sexes, sex-specific aspects of CO landscape are particularly poorly elucidated. Here, we conduct high-resolution CO mapping in maize. Our results show that CO numbers as well as their overall distribution are similar in male and female meioses. There are, nevertheless, dissimilarities at local scale. Male and female COs differ in their locations relative to transcription start sites in gene promoters and chromatin marks, including nucleosome occupancy and tri-methylation of lysine 4 of histone H3 (H3K4me3). Our data suggest that sex-specific factors not only affect male–female CO number disparities but also cause fine differences in CO positions. Differences between male and female CO landscapes indicate that recombination has distinct implications for population structure and gene evolution in male and in female meioses.

[1] Department of Horticultural Science, University of Minnesota, St. Paul, MN 55108, USA. [2] Section of Plant Biology, School of Integrative Plant Science, Cornell University, Ithaca, NY 14853, USA. [3] Bioinformatics Facility, Cornell University, Ithaca, NY 14853, USA. [4] Department of Plant Sciences, North Dakota State University, Fargo, ND 58102, USA. [5] National Center for Genome Resources, Santa Fe, NM 87505, USA. [6] USDA-ARS, Cereal Disease Laboratory, St. Paul, MN 55108, USA. [7] Present address: Eurofins BioDiagnostics, River Falls, WI 54022, USA. [8] Present address: National Maize Improvement Center, China Agricultural University, Beijing, China. These authors contributed equally: Penny M.A. Kianian, Minghui Wang. Correspondence and requests for materials should be addressed to P.M.A.K. (email: kiani002@umn.edu) or to S.F.K. (email: kiani001@umn.edu) or to W.P.P. (email: wp45@cornell.edu)

Although meiotic recombination is a key source of genetic variation, it does not affect the genome uniformly, as recombination events are unevenly distributed along chromosomes[1]. Factors affecting the location of recombination events are poorly understood. Meiotic recombination is initiated by formation of double-strand breaks (DSBs) in chromosomal DNA at the beginning of meiotic prophase[2]. DSB repair leads to two types of recombination products, crossovers (COs) and non-crossovers (NCOs)[3]. COs are reciprocal exchanges of chromosome arms, whereas NCOs are recombination products in which DSBs are repaired by DNA synthesis using as templates homologous DNA regions located either on sister chromatids or homologous chromosomes. In most species, the number of COs is only a small fraction of the DSB number[1,4]. In maize, there are ~200–500 DSBs per meiosis, as measured by the number of chromosomal foci of RAD51, a protein marker for DSB repair[5]. Yet, there are fewer than 20 COs[5]. The minimal as well as the maximal CO numbers per chromosome are strictly controlled[4]. For successful chromosome transmission to gametes each homologous chromosome pair requires at least one obligate CO. On the other hand, formation of multiple COs per chromosome is discouraged by CO interference, a phenomenon preventing formation of two COs close to each other. In most species, including maize, there are two distinct CO types[6]. Class I COs, which are the majority of COs in maize, are affected by interference, whereas class II COs, which constitute only ~15% of maize COs, are not[6]. However, even though class II COs do not exhibit interference themselves, studies in tomato have found interference between class II and class I COs[7].

Several factors are known to affect CO location, including chromatin state and local DNA sequence context[8–12]. Some of these factors vary among species. For example, in most vertebrates, locations of recombination events are associated with the presence of tri-methylation of lysine 4 of the H3 histone (H3K4me3) conveyed by the zinc-finger-containing SET domain protein PRDM9[13]. However, plants do not possess PRDM9 homologs. Most COs in plants are associated with genes[10–12,14,15]. The basis for this phenomenon may be the open chromatin environment found in genes, particularly gene promoters, which includes decreased nucleosome occupancy, DNA hypomethylation, and increased H3K4me3 levels[10–12]. In plants with large and complex genomes, such as grasses, there is also a tendency for a greater frequency of COs near chromosome ends, as compared to centromeric and pericentromeric regions[16,17]. Although the exact cause of this distribution is not known, distal regions of chromosomes are more gene-rich than pericentromeric regions in these species[18]. Furthermore, chromosome ends have been proposed to replicate their DNA earlier than pericentromeric regions, possibly resulting in their availability to recombination processes earlier than pericentromeric regions[16].

One of the least understood aspects of CO distribution are differences between male and female meioses. Disparities between the overall CO rates in the sexes have been observed in several species[8,19,20]. Interestingly, which sex exhibits a higher rate varies among taxa, even those closely related. For example, in human and mouse *Mus musculus castaneus*, females exhibit higher CO numbers. However, in *Arabidopsis* and another mouse subspecies, *M. m. musculus*, the opposite is true. In addition to CO number differences, variation in CO distribution between sexes has been described in humans and *Arabidopsis*[8,21]. In these taxa, the sex with the higher overall recombination rate shows a greater CO number in distal chromosome regions compared to the other sex. In *Arabidopsis*, these disparities are associated with different effects of the presence of transposable elements and protein-coding genes as well as GC content on the recombination landscape[8].

Although high-resolution CO maps have been generated in many species[10–12,14,15,17,21–23], studies of sex-specific CO patterns are few. Particularly, fine-scale differences in CO landscapes between the sexes, and how they relate to local genome and chromatin features, remain to be elucidated. Such studies are challenging, as examining CO patterns at high resolution limits the number of COs that can be analyzed, particularly in large-genome species.

To examine differences in CO patterns between male and female meioses, we combined Illumina sequencing with innovative bioinformatics strategies to generate high-resolution maps of COs in maize. Despite its relatively large genome size, maize is an excellent system for CO mapping due to the high number of intra-specific single nucleotide polymorphisms (SNPs)[24]. We found that CO numbers and distribution were fairly similar in male and female meioses. There were however, differences at local level. CO positions in the two sexes differed relative to open chromatin marks and COs located in gene promoter regions differed in their positions relative to transcription start sites (TSS). These observations suggest that even in species in which recombination patterns are similar in male and female meiosis, sex-specific factors affect CO positioning.

## Results

**CO numbers in male and female meioses in maize are similar**. To study recombination landscape, we created high-resolution maps of COs in male and female meioses in the B73 × Mo17 maize hybrid. We employed two backcross (BC$_1$) populations, B73 × (B73 × Mo17), which was used to examine male meiosis, and (B73 × Mo17) × B73, to examine female meiosis. To infer CO locations, plants were genotyped using Illumina sequencing to a coverage of at least ca. 1.5× (Supplementary Fig. 1). Analyses of sequence reads (see Methods) resulted in a total of 2.9 million SNPs distributed nearly genome-wide (Supplementary Fig. 2). The average marker density was one SNP per ~680 bp, whereas median density was one SNP per 44 bp. The discrepancy between these two numbers was largely due to the presence of regions entirely lacking DNA sequence polymorphisms, many of which corresponded to known locations of selective sweeps during maize domestication[24,25].

Overall, we found 1164 COs in 135 individuals genotyped in the male population and 1139 COs in 122 individuals in the female population, which translates to ~17.2 COs per male meiosis and ~18.7 COs per female meiosis. To identify CO positions, we devised a bioinformatic pipeline utilizing a Hidden Markov Model (HMM) analysis to correct for structural polymorphism between the B73 and Mo17 genomes (see Methods). The resulting marker spacing allowed us to determine CO locations with very high resolution (Supplementary Data 1), in many cases to the nearest SNP existing in the genome. Roughly half of the COs were mapped to within 2 kbp or less, 586 COs in male meiosis and 579 COs in female meiosis, for a total of 1165 COs. The CO numbers per meiosis were not statistically different between male and female according to the $\chi^2$ test. Furthermore, none of the differences between male and female CO numbers on individual chromosomes (Supplementary Table 1) were statistically significant. These data indicate that CO numbers in male and female meioses in maize are similar.

**Overall CO distribution is similar in male and female**. To compare recombination patterns in the two sexes on the global scale, we generated genetic maps using the CarthaGene package[26], and compared the ratio of genetic distances (cM) to physical distances (Mbp) in 1-Mbp-long intervals across the entire genome. Ratios in several intervals showed male–female

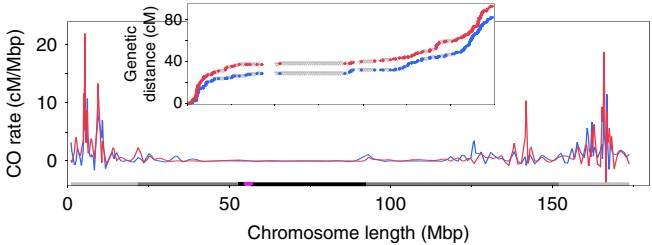

**Fig. 1** CO landscape of chromosome 7. See Supplementary Fig. 3 for other chromosomes. Blue = male COs, red = female COs. Colors on the X-axis indicate chromosome regions: purple = functional centromeres (=CENH3-binding regions[59]), black = centromere repeats, gray = pericentromeric regions, light gray = distal regions. Insets are cumulative genetic distances across entire chromosomes. Gray triangles in insets are supplemental SNPs added, where needed, to create 1 Mbp intervals. Figures were generated using MareyMap

differences, according to the likelihood ratio test. However, these differences were not statistically significant after applying corrections for multiple testing (Bonferroni, Benjamini–Hochberg, and Benjamini–Yekutieli). These results indicated that genome-wide CO distribution was similar in male and female meioses. Inspecting Marey maps, which serve to visualize the relationship between genetic and physical distances[27], also indicated similar CO distribution patterns in both sexes (Fig. 1, Supplementary Fig. 3).

In both sexes, there was an overall linear relationship between chromosome length and CO number (In male meiosis: $R^2 = 0.58$, $P < 0.0099$ according to the $F$-test; in female meiosis: $R^2 = 0.74$, $P < 0.001$ according to the $F$-test). However, the actual chromosome space where COs were present was relatively small, as only ~7% of 100-kbp-long windows across the genome contained COs. As expected, CO rates were higher in distal chromosome regions compared to centromeric and pericentromeric regions. However, as reported previously[14,15], at the very ends of chromosomes, COs were suppressed (Supplementary Table 2). These regions were on average ~3 Mbp-long, not including telomeric repeats, and were present in both male and female.

In addition to chromosome ends, there was a 17-Mbp long CO-repressed region on the short arm of chromosome 6 (Supplementary Fig. 3), which largely overlapped with the position of the nucleolus organizer region (NOR). However, the NOR was not entirely devoid of recombination. It contained eight COs, located between 17.1 Mbp to 22.0 Mbp from the chromosome end, although none of them overlapped with the actual rRNA gene arrays.

**Different local-scale CO distribution between male and female.** To compare male and female CO patterns at local scale, we identified CO hotspots, defined as regions 5 kbp in length exhibiting CO rates at least five-fold higher than the genome average[1]. We found 282 such sites in the male and 257 in the female (Supplementary Data 2). Only 66 of these sites (~14%) were shared by male and female. Furthermore, there was a negative correlation between CO rates at the hotspot sites between male and female meioses ($R = -0.712$). To validate this conclusion, we used a computer simulation similar to the one described by Taylor et al.[28], in which we employed a bootstrapping strategy with a 1000 replicates to compensate for the relatively low number of CO events available (see Methods). We found that the value of the correlation between male and female hotspots in the empirical population was not significantly different from these in resampled populations ($P = 0.4970$ according to the $Z$-test),

indicating that the empirical value was representative statistically and that increasing the population size would have little effect on our conclusion. We also examined in this way the top 10% strongest hotspots. In this group, the correlation between male and female CO activity was lower ($R = -0.276$) but the result was also representative statistically ($P = 0.4770$, $Z$-test).

As another way of examining CO distribution at a fine scale, we examined genes near CO sites using gene ontology (GO)[29]. Consistent with meiotic recombination being a major mechanism of gene evolution, studies in other species have uncovered that specific classes of genes, presumably those experiencing selective pressures to evolve faster, become CO hotspot sites. Not surprisingly, different gene classes are hotspots in different species. For example, in *Arabidopsis*, genes involved in disease resistance often show elevated CO levels whereas in potato, COs are associated with genes involved in regulation of transcription, transcription factor activity, and regulation of a cellular processes[30,31].

Of 573 unique genes identified within 10 kbp of the 586 CO sites mapped to within 2 kbp in male meiosis, 262 could be assigned a GO classification. In the female population, there were also 573 unique genes within 10 kbp from the 579 COs, of which 261 could be designated with a GO term. Fourteen GO terms were significantly enriched in the male; most of them were associated with phosphorylation-related processes (Supplementary Table 3). In the female, only two GO terms showed significant enrichment: oxidoreductase activity and cofactor binding. The GO terms significantly enriched in the male did not show noticeable enrichments in the female and vice-versa. Overall, the GO term analyses indicated that COs in male meiosis tend to be formed in different classes of genes than COs in female meiosis. Taken together, these data suggested that despite the overall similarity of their genome-wide distribution patterns, COs in the two sexes exhibit some location differences at local scale.

**CO-associated sequence motifs are similar in male and female.** To examine CO locations at an even finer scale, we searched for DNA sequence motifs associated with CO sites. Although functions of such motifs in plants are unclear, they have been found in several species, including *Arabidopsis*, maize, and tomato[10–12,15,32,33]. Using MEME[34], we identified three motifs most likely to appear at CO sites in male and female meioses, based on the subset of COs that were mapped to within 2 kbp. The motifs in the two sexes were nearly identical (Fig. 2). The two most frequent motifs were A/T polymers, which are thought to be associated with low nucleosome occupancy genome sites[10–12,35]. The third motif was a CG-rich trinucleotide repeat, which was similar to the DNA sequence motif associated with meiotic DSB sites in maize and the previously identified CO motifs in maize and *Arabidopsis*[10–12,33].

**Male and female show distinct CO locations in gene promoters.** The high-resolution dataset containing 1165 COs mapped to within 2 kbp or less allowed us to achieve high precision in examining the genomic context of CO sites. The distribution of the 1165 COs among chromosome regions mirrored the distribution of all COs (Supplementary Table 4), indicating that focusing on the COs mapped with higher resolution does not introduce bias. In agreement with previous studies in maize and *Arabidopsis*[10–12,15], we found that over 90% of the 1165 COs were within 10 kbp of a gene, which represents a statistically significant enrichment compared to a random distribution ($P = 9.201e^{-05}$ according to the Mann–Whitney–Wilcoxon test). This proportion was similar in male and female meioses (Fig. 3a). Also in both sexes, COs were generally excluded from close vicinities of

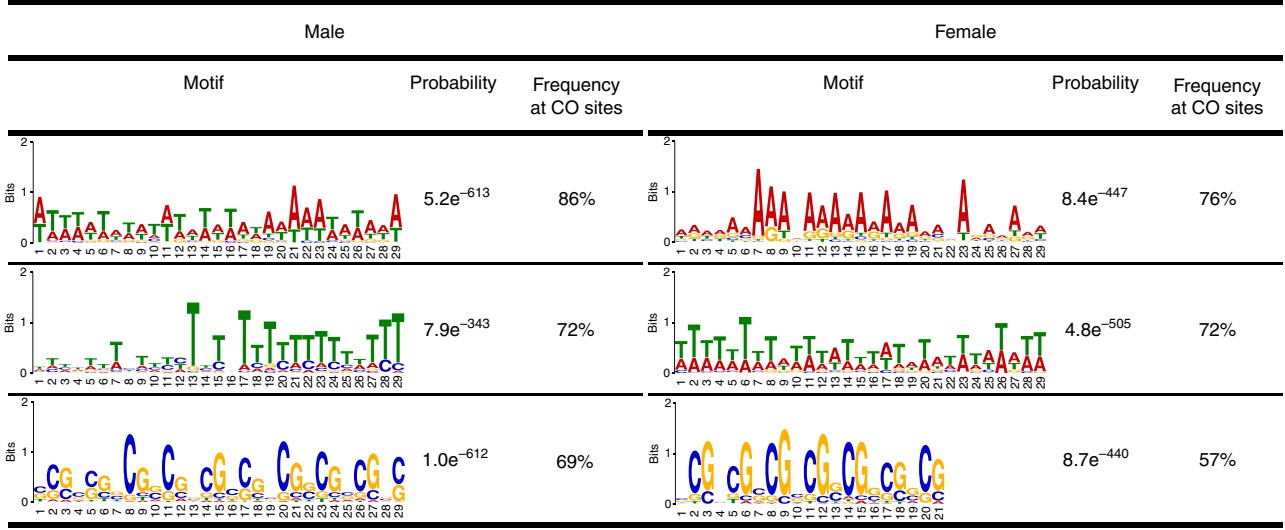

**Fig. 2** Comparison of DNA sequence motifs associated with CO sites in male and female meioses in the B73 × Mo17 hybrid. Probability values are log–likelihood ratios

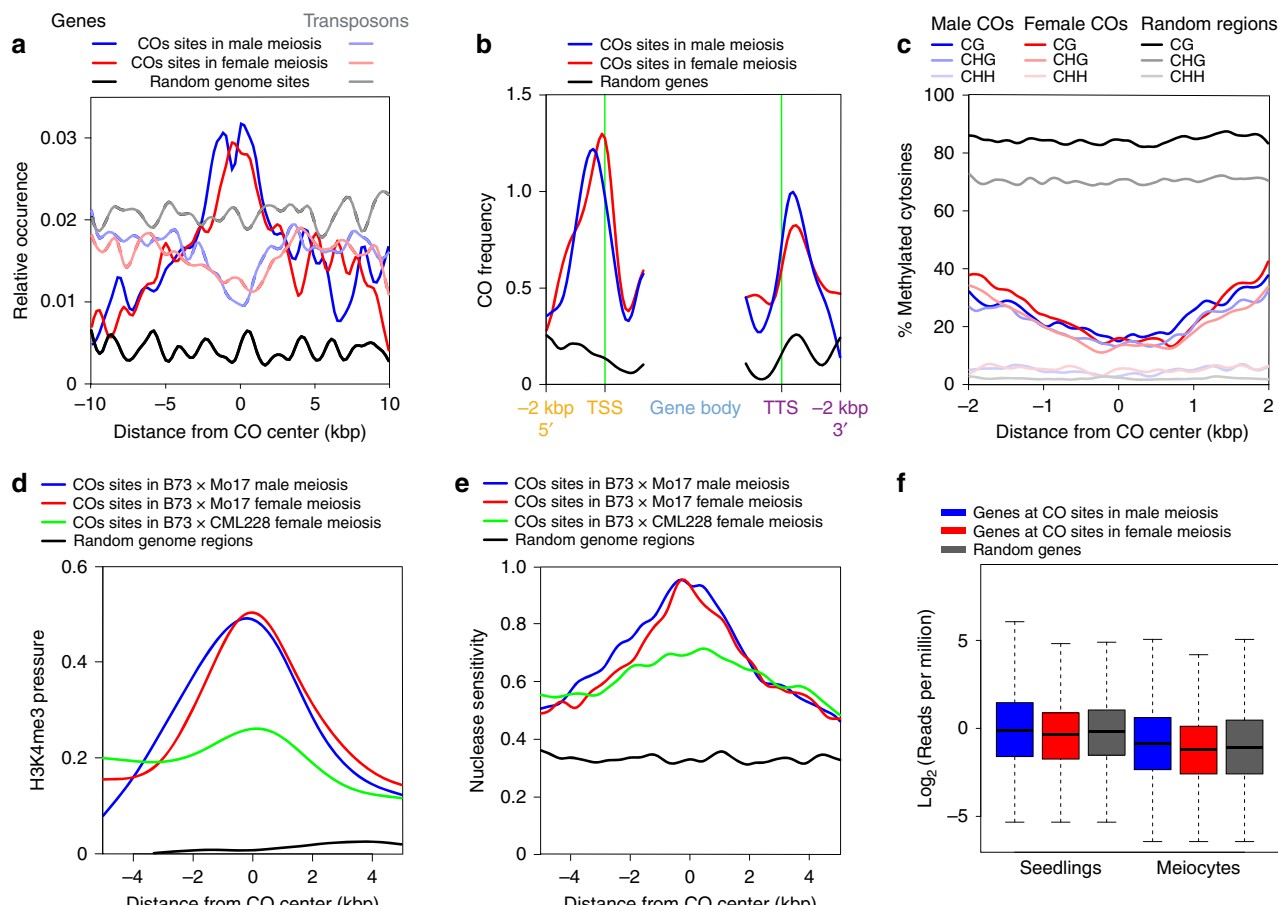

**Fig. 3** Relationship of COs sites to genome and chromatin features in male and female meioses. **a** Position of COs relative to TEs and protein coding genes. **b** Position of COs in the 5′ and 3′ regions of genes. **c** DNA methylation patterns around CO sites at CG, CHG, and CHH sites. **d** H3K4me3 levels of CO sites in male and female meioses. **e** Nucleosome occupancy of CO sites in male and female meioses measured using micrococcal nuclease sensitivity. **f** Boxplot showing transcription levels of CO-containing genes in male meiocytes and seedlings. Center lines of boxes represent median expression levels, bounds of boxes designate expression levels of the first ($Q1$) and third ($Q3$) quartiles from top, lower whiskers are $Q1 – 1.5 \times IQR$, and upper whiskers are $Q3 – 1.5 \times IQR$, where $IQR = Q3 – Q1$

transposable elements (TEs) (Fig. 3a). This finding is in contrast to Giraut et al.[8], who discovered a positive correlation of CO locations with TE presence in female meiosis but not in male meiosis in *Arabidopsis*. These patterns may represent dissimilarities between maize and *Arabidopsis* recombination landscapes. Alternatively, the outcomes of the two experiments could have been confounded by differences in CO mapping resolution, which was relatively low in the *Arabidopsis* study, combined with the higher gene density in *Arabidopsis* than in maize.

Within genes, ~30% of COs were located within 2 kbp upstream from transcription start sites (TSS) and ~20% were within 2 kbp downstream from transcription termination sites (TTS). Even though these proportions were similar in both sexes, detailed patterns of CO distribution within genes significantly differed between male and female meioses ($P = 0.0021$ according to the Kolmogorov–Smirnov test). In particular, the male and female CO density peaks did not converge in TSS regions ($P = 0.0088$ according to the $Z$-test). Female COs increased next to the TSS whereas in the male, the CO peak was ~400 bp upstream from the TSS (Fig. 3b). On the other hand, COs located in the 3′ of genes showed peaks at the same location in male and female meioses, at ~400 bp downstream from the TTS (Fig. 3b).

**COs locations show differences relative to open chromatin**. We also used the set of 1165 COs mapped to within 2 kbp to examine chromatin features underlying CO presence. We found that CG and CHG methylation levels at male and female CO sites were similar and substantially lower than the average CG and CHG genome methylation level (Fig. 3c). CHH methylation levels, although also alike at male and female CO sites, were similar to the genome average of about 10% of methylated cytosines (Fig. 3c).

In contrast to DNA methylation, statistically significant differences were detected in the position of male and female COs relative to two marks of active chromatin, H3K4me3 and nucleosome occupancy. Genes harboring COs in male and female meioses exhibited similar H3K4me3 patterns and both showed higher H3K4me3 levels than an average gene (Supplementary Fig. 4). However, in the female, H4K4me3 levels increased at CO sites, whereas in the male they peaked ~250 bp upstream from CO sites (Fig. 3d). The difference in the locations of the two peaks were statistically significant ($P = 0$ according to the $Z$-test). Genes at male and female CO sites did not differ in nucleosome occupancy patterns, measured by micrococcal nuclease digestion, and showed lower nucleosome occupancy than an average gene (Supplementary Fig. 4). In the female, the CO peak was ~300 bp downstream from the middle of the nucleosome-depleted region. In contrast, in the male, there were two peaks of nucleosome depletion, located ~600 bp apart from each other (Fig. 3e). Positions of the two male peaks were statistically distinct ($P = 5.166824e^{-245}$, $Z$-test). The position of the left male peak was also statistically distinct from the position of the female peak ($P = 1.27606e^{-40}$, $Z$-test), although the distance between the two peaks was only ~45 bp.

As both transcription and recombination require open chromatin[36], we examined the relationship between CO sites and gene expression during meiotic prophase. Transcription levels were similar for genes at male and female CO sites, and genes at both types of sites showed expression levels close to the average for all meiotically expressed genes (Fig. 3f). We found that genes located at CO sites were also expressed in seedlings, and the expression levels in seedlings and meiocytes were similar. Overall, there was no evidence of a link between CO sites and gene transcription patterns.

For the chromatin features and transcriptome analyses, we used data generated from male meiocytes (H3K4me3 and transcriptome[37,38]) and anthers (nucleosome occupancy), as conducting chromatin analyses of female reproductive cells in plants is not feasible with current technology. In previously published studies, chromatin features associated with CO presence, including DNA methylation, H3K4me3, and nucleosome occupancy, were identified using chromatin data from somatic tissues[10–12,15], suggesting that chromatin landmarks exhibit stability throughout the plant life cycle. To further examine this issue, we compared H3K4me3 and nucleosome occupancy at male and female CO sites using chromatin data from male reproductive tissues and seedlings. We found that the H3K4me3 and nucleosome occupancy patterns in reproductive and somatic cells were similar and exhibited peaks at the same locations (Supplementary Fig. 5). These results further implied life-cycle stability of chromatin patterns at CO sites.

Taken together, our analyses indicated that both male and female COs sites were associated with decreased levels of DNA methylation and nucleosome occupancy, as well as higher H3K4me3 levels. However, male and female CO sites exhibited distinct features relative to nucleosome occupancy and H3K4me3.

**Chromatin features at CO sites in diverse germplasm**. To examine whether the similarities and differences between male and female COs persist in different genetic backgrounds, we mapped COs in female meiosis in a hybrid between B73 and a tropical inbred line CML228. Mapping CML228 sequence reads to the B73 reference genome suggested a higher level of structural polymorphisms between the CML228 and B73 genomes than between those of B73 and Mo17. Because an assembled reference genome sequence for CML228 is not available, this situation prompted us to adjust the genotyping pipeline to avoid read misalignments that would lead to spurious CO identification[38]. Using this conservative approach, we found 790,000 SNPs in between B73 and CML228, allowing us to identify 765 COs (19.1 COs/meiosis), of which 195 were mapped to within 2 kbp or less. Although this number may be too low to investigate CO distribution at specific genome sites, it is sufficient, based on previously published studies[11,14], to conduct analyses of CO positions relative to chromatin features.

To investigate whether CO patterns in the B73 × CML228 hybrid were similar to those in B73 × Mo17, we analyzed CO locations relative to H3K4me3 and nucleosome occupancy patterns. We found that the H3K4me3 peak at CO sites in B73 × CML228 was at the same location as for the B73 × Mo17 female COs, even though the H3K4me3 level at CO sites in B73 × CML228 was lower than that in B73 × Mo17 (Fig. 3d). The H3K4me3 patterns were essentially the same regardless of if we used H3K4me3 data for B73 plants or for CML228 plants (Supplementary Fig. 6).

Nucleosome occupancy levels at CO sites in the B73 × CML228 population were also less pronounced than those in B73 × Mo17. However, in contrast to the H3K4me3 patterns, the sites of nucleosome occupancy peaks in B73 × CML228 did not overlap with the peaks observed in female meiosis in B73 × Mo17.

Overall, the comparison of the B73 × CML228 and B73 × Mo17 CO data suggested that some chromatin features at CO sites in male vs. female meioses may be universal in maize (H3K4me3) whereas others (nucleosome occupancy) may pertain to specific crosses.

**Chromatin features in regions of CO suppression**. Finding significant differences in the location of chromatin marks relative to COs in male vs. female meioses prompted us to investigate how

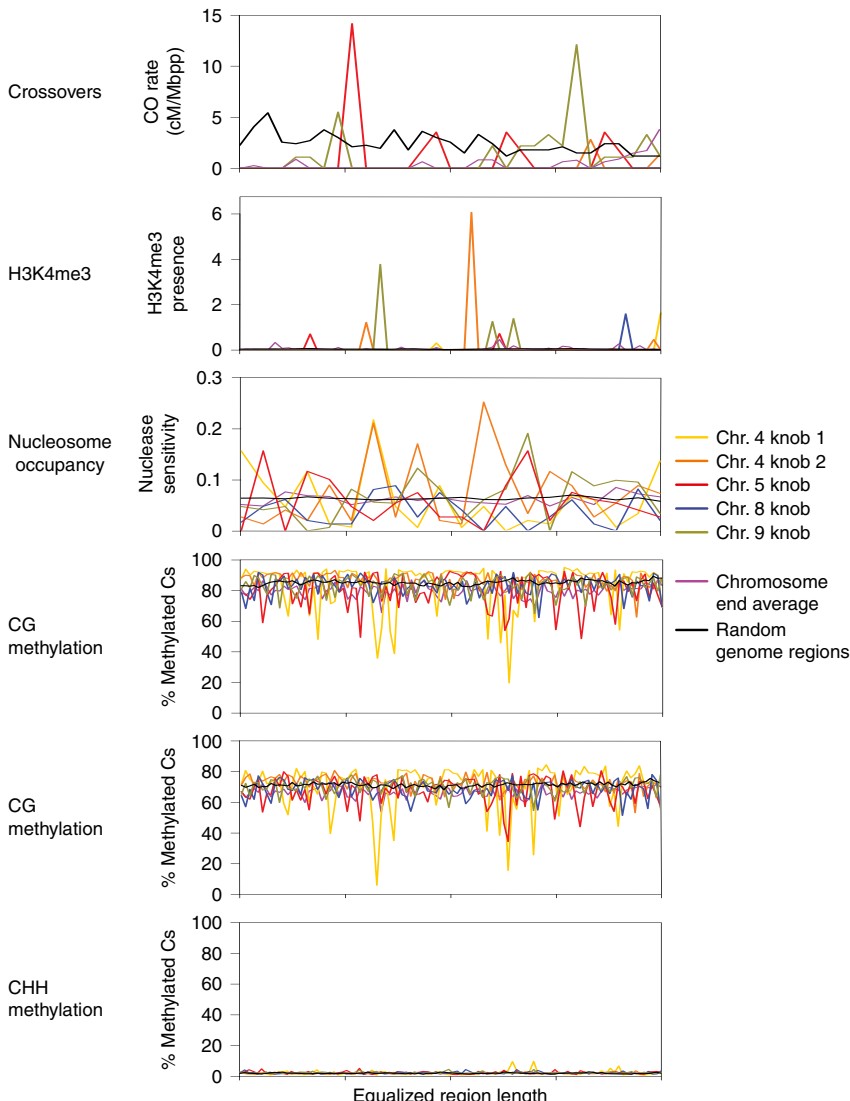

**Fig. 4** Recombination and chromatin features at knobs and chromosome ends. All regions were equalized to the same size and divided into 100 bins for analysis, except nucleosome occupancy, which was divided into 30 bins to improve readability of the graph. The five knobs mapped in B73 are shown. The sixth knob present in B73 and the single knob present in Mo17 are not anchored to the maize genome scaffold and were not analyzed

strongly specific chromatin features determine CO landscape. To do this, we used the B73 × Mo17 CO maps to examine two regions exhibiting CO suppression relative to immediately adjacent sites, heterochromatic knobs and chromosome ends. Knobs are stretches of highly repetitive DNA, consisting predominantly of two sequence types, the 180 bp repeat and the TR-1 repeat[39], and are thus similar in their overall DNA sequence composition. They are, however, present at different chromosome locations[40,41]. In contrast, chromosome ends have the same position but are more variable in sequence.

Despite their repetitive makeup, knobs have varied effects on recombination[17,41,42]. We found CO suppression in three of the five knobs included in the B73 reference genome (Fig. 4), whereas two knobs showed CO rates higher than the genome average. H3K4me3 levels varied among knobs and, surprisingly for repetitive DNA regions, were overall higher than the genome average (Fig. 4). On the other hand, nucleosome occupancy patterns were largely similar at all knobs (Fig. 4) and the average nucleosome occupancy level at knobs was similar to the genome average. DNA methylation patterns at CG and CHG sites varied among knobs (Fig. 4). However, on average, DNA methylation

levels at knobs were similar to the genome average and significantly higher than DNA methylation levels at average CO sites (Fig. 3c). Overall, neither of the chromatin marks examined showed correlation to CO patterns in the knob regions.

In contrast to knobs, chromosome ends always lacked COs (Supplement Table 2, Fig. 4). In yeast, CO absence near telomeres has been ascribed to the heterochromatic structure of the subtelomeric regions[43]. However, we did not find evidence that chromosome ends in maize were exceptionally heterochromatic. H3K4me3 and nucleosome occupancy at chromosome ends were comparable to the rest of the genome (Fig. 4). CG and CHG methylation levels, although variable (Fig. 4), were overall slightly lower than the genome average. On the whole, these results did not point to a specific chromatin modification pattern responsible for the absence of COs at chromosome ends.

## Discussion

A high-resolution map of COs in the B73 × Mo17 hybrid allowed us to compare recombination landscapes in male and female meioses. We found that the CO number and their overall

distribution along chromosomes were fairly similar in the two sexes. Previous studies in maize uncovered a strong link between the CO number and their chromosomal position[5]. Thus, the fact that the overall CO numbers in male and female meioses are roughly the same, unlike in human, mouse, or Arabidopsis[8,21,44,45], allows for uncoupling the effect of the sex on CO landscape from the effect of CO number.

Despite the overall similarities in CO distribution on chromosomes, there were differences at local level. First, male and female COs were found near different classes of genes and the two sexes differed in the location of the majority of CO hotspots. Second, COs located in gene promoters differed in their position relative to the TSS. Third, H3K4me3, and to some extent, nucleosome occupancy patterns differed at male and female CO sites.

The male–female differences could be byproducts of chromatin landscape differences between male and female meiocytes. However, the fact that CO sites exhibited similar chromatin patterns in male meiocytes and somatic tissues argues against this possibility. A more likely reason is existence of sex-specific mechanisms explicitly affecting CO location. The distances between H3K4me3 peaks at male and female COs and between the two nucleosome occupancy peaks at male COs seem to roughly correspond to multiples of the DNA length taken by nucleosomes. Although the significance of this observation is unclear, it would be tempting to speculate that the peaks of chromatin openness at male and female COs could differ by a specific number of nucleosome positions.

Analyzing CO locations in the B73 × CML228 hybrid, whose two parents were more diverse than B73 and Mo17, we found H3K4me3 patterns at CO sites that were fairly similar to those in female meiosis in B73 × Mo17, suggesting evolutionary conservation. On the other hand, nucleosome occupancy patterns in B73 × CML228 female meiosis were different from those in B73 × Mo17 female meiosis. This observation implies that the effect of nucleosome occupancy on recombination landscape may be under a more complex genetic control, rather than being just sex-dependent.

Taken together, our data uncovered that even in species in which male and female CO landscapes are similar, fine-scale CO pattern differences may exist between the sexes. Mechanisms responsible for these differences may also operate in species with substantial male–female differences, but the effects of these differences may be overshadowed by the effects of CO number differences. We hypothesize that mechanisms regulating CO positions relative to chromatin features such as H3K4me3 are at the core of CO landscape differences between sexes.

Differences in recombination landscapes between male and female meioses are expected to result in distinct patterns of genetic diversity created by the two sexes. Recombination generates new allelic combinations as well as novel alleles, as most COs are within genes. Hence, differences in diversity patterns created by male and female meioses may have implications for population structure and evolution of specific genes. These implications could be particularly important in open pollinated species, in which products of male meiosis (pollen) can travel for extended distances whereas female meiosis products do not, as seed dispersal (which is the travel mode of female meiosis products) is more limited than pollen dispersal[46]. On a practical level, the knowledge of different diversity patterns created in male vs. female meioses could inform the direction of crosses in genetics and breeding.

Male and female meioses in maize both exhibited U-shaped CO distribution patterns along chromosomes, with more COs located near chromosome ends than in centromeric/pericentromeric regions. Typically, a role for chromatin structure in determining CO location has been invoked to explain these patterns, particularly since the CO-poor pericentromeric regions are gene-poor and contain more TE than distal regions[15]. However, many chromatin modification patterns in maize, such as DNA methylation and H3K4me3[47], do not trail the strongly U-shaped CO distribution. Therefore, the role of chromatin structure in explaining CO landscape in maize and other large-genome species requires further studies.

CO sites, overall, exhibited DNA methylation levels significantly lower than the genome average. However, some knobs showed elevated CO rates compared to the genome average, despite exhibiting high DNA methylation levels. Chromosome ends, on the other hand, completely lacked COs, even though they did not show elevated DNA methylation compared to the rest of the genome. Previous studies in Arabidopsis plants with artificially altered DNA methylation patterns, have pointed to a complex relationship between DNA methylation and recombination. Global reduction in DNA methylation levels altered recombination patterns in regions that were heterochromatic in wild type as well as in regions that were already euchromatic[48–51]. This finding suggested that DNA hypomethylation affected recombination by influencing chromatin structure chromosome-wide[52], rather than enabling specific demethylated sites to become recombination hotspots. On the other hand, increasing methylation levels locally was sufficient to silence recombination hotspots[52]. However, our observations of CO presence in heterochromatic knob regions in maize suggest that the effect of DNA hypermethylation is also indirect and proceeds via an overall change in chromatin structure. Collectively, these data suggest that DNA hypomethylation is required but not sufficient for CO formation.

The role of H3K4me3 in designating CO sites may be even less direct than that of DNA methylation. Our data showed sex-specific differences in H3K4me3 patterns at CO sites in male vs. female meioses. In knob regions, CO patterns were independent from H3K4me3 patterns. Thus, the relationship between CO location and H3K4me3 may be by virtue of most COs being located in gene promoters. A similar explanation was proposed in yeast[53]. This possibility is also consistent with DSB hotspots, only few of which are in gene promoters, not generally showing elevated levels of H3K4me3[33].

In contrast to DNA methylation and H3K4me3, the overall pattern of nucleosome occupancy along maize chromosomes is generally similar to CO distribution[54]. Furthermore, our data, and data from previous studies in maize, Arabidopsis, and other species indicate that recombination events are formed in nucleosome-free regions[10,11,33,54,55]. However, while this characteristic may be necessary for CO formation, it is not sufficient, as the number of nucleosome-free sites far exceeds the number of recombination hotspots[33,54]. This is also true for nucleosome-free regions in gene promoters, where many maize COs are formed. As many as 16,000 genes, which equals to roughly one-half of all genes in maize, are expressed in male meiocytes at the time when meiotic recombination takes place[38] and their promoters are likely to exhibit open chromatin structure. Interestingly, the peaks of nucleosome-free regions in gene promoters were at different positions relative to CO sites in male and female meioses. This observation may indicate existence of either sex-specific factors controlling CO location or factors specific to the different classes of genes that were CO sites in the two sexes. It also suggests that a mechanism of CO preference for nucleosome-depleted sites may be complex.

Overall, our analyses did not identify a single chromatin feature defining locations of CO sites in maize. It may be a combination of chromatin features that collectively convey a specific open chromatin pattern necessary for a site to become a CO

location. Alternatively, it is possible that some yet undefined chromatin characteristics control CO location in a more definite manner. Detailed studies of chromatin conformation at CO sites may shed light on this issue.

## Methods

**Genotyping**. DNA was extracted from leaf tissue following instructions of the QIAGEN DNeasy Plant Mini kit (Hilden, Germany), quantified, and sonicated (Covaris, Woburn, MA, USA) to approximately 200 bp. Illumina sequencing libraries were prepared using the NEBNext DNA Library Prep Master Mix (New England Biolabs, Ipswich, MA, USA) following manufacturer's instructions. AMPure XP beads (Beckman Coulter Inc., Pasadena, CA) were used for size selection, and NEBNext Multiplex Oligos Index Primer Set 1 (New England Biolabs) was used for end labeling. The libraries were sequenced on an Illumina HiSeq instrument to produce $2 \times 100$ bp paired-end reads. Base calling and initial data processing were performed using the standard Illumina protocol.

To genotype the B73 × Mo17 populations, Illumina sequence reads passing quality control were aligned to the maize B73 reference genome scaffold using the BWA mem protocol[56]. Reads with sequence quality <30, reads with mapping quality scores <5, and reads that mapped to multiple genome locations were discarded. Duplicate reads were removed by PicardTools (version 1.98) and SNP calling was performed using the Genome Analysis Toolkit (GATK, ver. 3.0-0; http://www.broadinstitute.org/gatk). Variant detection and genotype calling were performed with the UnifiedGenotyper tool in GATK using default parameters except that the parameter -mbq was set at 20. Variants with DP values <120 and >2000, and FSfilter values >60 as well as HaplotypeScore values >13 were removed using the VariantFiltration tool.

To produce the B73-Mo17 SNP set, the Illumina reads were compared to an Illumina-generated $17 \times$ whole-genome sequence of Mo17 (Genbank accession number SRA051245). Only SNPs exhibiting allelic frequencies in our data sets from 0.1 to 0.4 were retained. Then, a linkage disequilibrium test was performed in 10-Mbp intervals, and only SNPs with medium $P$ value <0.05 were kept. Finally, spurious SNPs caused by copy number differences in B73 vs. Mo17 were discarded by analyzing B73 and Mo17 sequence coverage patterns.

To genotype the B73 × CML228 population, we followed the same protocol as for B73 × Mo17 with a few modifications. Illumina sequence reads were aligned to the version 4 of the maize genome scaffold and filtered to only retain properly paired reads with mapping quality >60, allowing up to three mismatches. Variants with DP values <100 and >450 were also removed. For SNP calling, we used an Illumina sequence of CML228 from a species-wide maize haplotype analysis (Genbank accession number PRJNA399729), supplemented with our own whole-genome Illumina sequence to achieve a combined coverage of 17.3×. The combined sequence was used to compare reads from the B73 × CML228 population. HaplotypeCaller implemented in Sentieon DNAseq (ver. 201711.01) software (Sentieon Inc., Mountain View, CA, USA) was employed to identify SNPs using default settings, except that --call_conf and --emit_conf were both set at 30. Only SNPs exhibiting allelic frequencies in our data set from 0.13 to 0.37 were retained.

**CO inference**. To determine CO positions, we first calculated the frequency of Mo17 or CML228 alleles in non-overlapping 200 kbp bins across the genome. Then, the genome was partitioned into segments of bins with different genotypes (B73 homozygote and heterozygote) and genotype breakpoints were identified by the mean-shift method implemented in the cumSeg package in R (Supplementary Fig. 7). Subsequently, all SNPs in the 10-Mbp region centered on each breakpoint of interest were used in a Hidden Markov Model (HMM) analysis[57] to compute the posterior probability that a specific genome region is either homozygous for B73 or heterozygous.

**CO landscape analyses**. CO data were analyzed using CarthaGene[26] two- point linkage with the Haldene function to calculate genetic distance. Some chromosome regions had no SNPs resulting in intervals larger than 1 Mbp. To create 1-Mbp intervals, we added appropriately spaced supplemental sites in regions where SNPs were not available. The genetic to sequence map relationship was analyzed with MareyMap[58] using spline regression. Chromosome features were overlaid on the maize genome sequence scaffold[18]. Centromere locations were from Wolfgruber et al.[59] and Schneider et al.[60]. Knob locations were based on genetic mapping by Ghaffari et al.[61,62].

**Nucleosome occupancy mapping**. Male flowers at the zygotene stage of prophase I were fixed in 1% formaldehyde for 10 min, and then quenched in 0.125 mM glycine for 5 min. About 1.5 g of the fixed flower tissue was ground into fine powder in liquid nitrogen and homogenized in Chromatin Extraction Buffer A (10 mM Tris–HCl pH 8.0, 0.4 mM sucrose, 10 mM MgCl₂, 1 mM PMSF, 5 mM β-mercaptoethanol, and 1 tablet of cOmplete Protease Inhibitor Cocktail (Roche Applied Science, Indianapolis, IN, USA) per 50 mL of buffer) for 20 min at 4 °C with gentle shaking. The suspension was filtered into a new 50 ml conical tube through two layers of Miracloth (EMD Millipore, Billerica, MA, USA), placed in a plastic funnel and centrifuged at 4000 rpm for 20 min at 4 °C. The pellet was

resuspended in 1 mL of Extraction Buffer B (10 mM Tris–HCl pH 8.0, 0.25 M sucrose, 10 mM MgCl₂, 1% (v/v) Triton X-100, 1 mM PMSF, 5 mM β-mercaptoethanol, and 1 tablet of cOmplete Protease Inhibitor Cocktail (Roche Applied Science) per 50 mL of buffer), and centrifuged at 14,000 rpm for 10 min. The pellet was resuspended in 500 μL of Extraction Buffer C (10 mM Tris–HCl pH 8.0, 1.7 M sucrose, 2 mM MgCl₂, 0.15% (v/v) Triton X-100, 1 mM PMSF, 5 mM β-mercaptoethanol, and 1 tablet of cOmplete Protease Inhibitor Cocktail (Roche Applied Science) per 50 mL of buffer), placed on top of 500 μL of Extraction Buffer C cushion, and centrifuged at 14,000 rpm for 1 h at 4 °C. The pellet containing nuclei was resuspended in 500 μL of Digestion Buffer (50 mM Tris–HCl pH 8.0, 5 mM CaCl₂, 0.1 mM PMSF, and 1 tablet of cOmplete Protease Inhibitor Cocktail (Roche Applied Science) per 50 mL of buffer). Samples were sonicated for 5 s and treated with 1 U μL⁻¹ of micrococcal nuclease (NEB, Ipswich, MA, USA) for 10 min at room temperature. The reaction was stopped with 10 mM EDTA. Chromatin was treated with RNase for 2 h at 37 °C, incubated with Proteinase K for 2 h at 45 °C, and decrosslinked at 65 °C for 8 h. DNA was purified using MinElute PCR Purification Columns (Qiagen, Hilden, Germany). The purified DNA was separated in a 2%(w/v) agarose gel and fragments ~150 bp in size were recovered. Approximately 100 ng of mononucleosome DNA was used for Illumina library construction. As a control, randomly fragmented chromatin was prepared by sonication to produce 200–500 bp fragments.

**H3K4me3 chromatin immunoprecipitation**. Male meiocytes in leptotene and zygotene were collected using the capillary collection of meiocytes (CCM) method[37]. Maize tassels were fixed in 10 mM Tris–HCl pH 8.0, 0.25 M sucrose, 10 mM MgCl₂, and 1% (v/v) formaldehyde for 10 min and then quenched for 5 min by adding 1/10 volume of 1.25 mM glycine. They were then washed twice in 1× PBS and stored in 1× PBS in 4 °C until use. Anthers were dissected, squashed in a drop of 1× PBS. Clusters of meiocytes were aspirated using a microcapillary. A purification step was performed by transferring the collected meiocytes into a fresh 1× PBS drop. About 30,000 meiocytes were pooled and pelleted by centrifugation at 3000 rpm for 2 min. The pellet was resuspended in 100 μL of Extraction Buffer A (see Nucleosome occupancy mapping) and the meiocytes were ground using a microtube pestle, followed by addition of another 900 μL of Extraction Buffer A. The sample was incubated on ice for 30 min with periodical vortexing and filtered through Miracloth (EMD Millipore, Billerica, MA, USA) pre-wetted Extraction Buffer B (see Nucleosome occupancy mapping) into a new tube. A cut 1000 μL pipet tip was used to transfer the meiocyte suspension. The sample was then centrifuged at 6000 rpm for 20 min in 4 °C. The pellet was resuspended in 1 mL of Extraction Buffer B, transferred on top of 300 μL of Extraction Buffer C (see Nucleosome occupancy mapping) and centrifuged at 13,000 rpm for 1 h at 4 °C. The nuclei pellet was resuspended in 50 μL of Nuclei Lysis Buffer (see Nucleosome occupancy mapping). The extracted chromatin was sonicated using the Bioruptor 200-UCD (Diagenode, Denville, NJ, USA) to produce fragments 100–300 bp in length. ChIP was performed using the MAGnify Chromatin Immunoprecipitation System (Invitrogen, Carlsbad, CA, USA) with 10 μL of rabbit polyclonal anti-trimethyl-histone (Lys4) antibody (EMD Millipore, Billerica, MA, USA, Catalog #07-473)[37]. Standard Illumina protocols were used for library construction and sequencing.

**Statistical analyses**. To analyze genome-wide CO distribution, the likelihood ratio test was used to compare CO numbers in male and female in 1-Mbp-long intervals to those of randomly selected 10,000 genomic regions of the same size with a null hypothesis that the recombination rate = 0. Differences were called when the test value exceeded the 95% confidence interval.

To examine differences in CO hotspot distribution, we used a bootstrapping strategy, in which we generated 1000 replicates by randomly sampling with replacement from the 473 genomic sites consisting of empirical male and female CO hotspots. The $P$ value was calculated by counting frequency of correlation coefficients that were higher than the empirical $R$ and dividing it by 1000.

To analyze CO location relative to gene elements and chromatin features, we first examined their overlap with BEDtools (ver. 2.26.0; http://code.google.com/p/bedtools/), using the mid-point of the CO interval as CO location. Differences in distribution between male and female COs were also tested using a bootstrapping approach. CO locations were randomly selected 1000 times with replacement, and mean differences between male and female CO distribution were calculated per bootstrap sample.

GO analyses[29] were performed against Version 5a of maize genome IDs using the Fisher's test with Bonferroni correction.

**Data availability**. Illumina sequence reads from genotyping the B73 × Mo17 populations and the B73 × CML228 populations are available from Genbank under accession code PRJNA336121 Illumina sequence reads from H3K4me3 ChIP in the B73 inbred are available from Genbank under accession code PRJNA185817 Illumina sequence reads from H3K4me3 ChIP in the CML228 inbred are available from Genbank under accession code PRJNA451000 and Illumina sequence reads from nucleosome occupancy mapping in the B73 inbred are available from Genbank under accession code PRJNA328990.

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

## Acknowledgements

We would like to thank A. Peckrul and J. Hegstad for technical assistance, N. Springer for discussion, and T. Pawlowska for discussion and advice on computer modeling. This research was supported by grants from the U.S. National Science Foundation (IOS-1025881 and IOS-1546792)

## Author contributions

P.M.A.K., M.W., Q.S., J.P., C.C., S.F.K., and W.P.P. designed the project. P.M.A.K., K.S., and F.G. conducted plant genotyping. Y.H. generated mapping populations and contributed nucleosome occupancy data. S.D.S., A.S., J.M., and C.C. contributed H3K4me3 data. P.M.A.K. and M.W. conducted data analyses. P.M.A.K., M.W., and W.P.P. wrote the manuscript. P.M.A.K., M.W., Q.S., S.D.S., C.C., S.F.K., and W.P.P. edited the manuscript.

## Additional information

**Competing interests:** The authors declare no competing interests.

