## [Peer Review File · Nature Communications]

Reviewers' comments:

Reviewer #1 (Remarks to the Author):

This paper carefully examines numerous different aspects of recombination patterns in maize, looking separately at male and female using opposite backcrosses. The analyses are all thoughtful, careful, and well-described. I appreciate the careful detail in the analytical descriptions. I do not work in maize (I study recombination in humans), so I am not sure how important these insights are in maize, but this is a nice contribution to the overall picture of male vs. female recombination across species.

My only concern is that the paper is written to emphasize the few areas in which the authors found male/female differences. From my point of view the big story is how similar male and female are, especially as compared to other species that show very big male/female differences in overall recombination count, gross locations of recombination, etc. I suspect that the authors felt like differences are higher-impact than similarities, but the analyses that show differences between males and females seem to me to be the weakest ones, with most statistical tests separately comparing males and females to some "norm" rather than directly comparing them to each other. The claim that GO analysis shows "different classes of genes" near male and female crossovers seems particularly overblown to me. I would not insist that the authors revise the manuscript based on these comments, but I would encourage them to think about whether the bigger story here is the differences or the similarities.

Reviewer #2 (Remarks to the Author):

Meiosis is a fundamental biological process that plays a crucial role in all sexually reproducing organisms. In the case of higher plants it underlies the development and breeding of new crops with improved traits. Homologous recombination during meiosis results in the formation of genetic crossovers. Crossover formation creates new combinations of alleles that can be exploited by plant breeders. In plants, notably cereals and many other organisms the distribution of crossovers along the chromosomes is non-uniform. Some regions, generally but not always distal to the centromeric region, exhibit crossover formation whereas others notably proximal regions are almost devoid of crossovers. Also overall crossover frequency can vary between males and females. Although these aspects of crossover formation have been known for many years, understanding the chromosomal features that influence them is relatively poorly

understood particularly in non-model species. Elucidating their nature is particularly important in the case of crop species as the non-uniform crossover distribution is an important impediment to breeders. Understanding these factors may well provide the basis to develop approaches to modify crossover distribution.

The research described in this manuscript by Kianian et al represents an important contribution to analyzing crossover formation in maize. It is well executed and carefully analyzed, thereby allowing the authors to correlate the distribution of crossovers with chromosomal features in female and male meiosis. They report both similarities and differences to those observed in previous studies conducted in *Arabidopsis*.

In contrast to *Arabidopsis* and mammals they did not observe any significant difference in crossover frequency between male and female meiosis. However, they provide evidence to indicate that the distribution of crossovers in female and males is distinct. At a global level they find that crossover distribution is similar but when crossover hotspots are analysed in more detail they find little overlap. In both cases they find a relationship between crossovers and regions of open chromatin with low nucleosome occupancy and the H3K4me3 mark. As in *Arabidopsis* they observe a preference for transcription start sites and transcription termination sites.

However intriguingly, the exact relationship with these features is slightly different in males and females. In males crossovers are enriched in association with a different group of genes (based on GO terms) than those in females.

Together these observations provide an interesting insight into the relationship between the sites of crossovers and chromatin features in maize. The limitation of the study is that based on their results they are unable to really provide insight into the key causal factor(s) or mechanisms that explain their data. For instance is there any functional significance in the fact that crossover hotspots in males are enriched in association with a different group of genes than hotspots in females? Despite this limitation the study certainly advances our knowledge of how crossovers are distributed within the genome of a major crop species and raises questions to address in further studies.

Reviewer #3 (Remarks to the Author):

In this manuscript Kianian and colleagues compare CO frequency and distribution in female and male maize meiosis and correlate those patterns with (epi)genomic features such as genes, TSS, DNA methylation, H3K4me3 and nucleosome density. Understanding the genetic mechanisms that control CO frequency and distribution is a central problem in the meiotic recombination field and is thus appropriate for the scope of *Nature Communications*.

Major Concerns

1. The study is entirely descriptive. This kind of genomic profiling is an appropriate starting place for querying sex differences in CO control, but given the rich genetic resources available in maize follow-on experimentation to test the emerging hypotheses is within the expectations of the maize genetics community.
2. All of the conclusions are based on crosses between B73 and Mo17. It is entirely possible that the correlations that the authors describe are idiosyncratic to that cross and do not represent maize broadly, much less other plant species. This significantly reduces the importance of the findings for the broad genetics community.
3. The study surpasses many similar prior studies in other plants that relied on comparing somatic epigenetic profiles to meiotic CO frequencies/distributions. In this study the authors made the effort to provide a more direct comparison to epigenetic profiles from male meiocytes – this was a real strength for the paper. Even so, the authors go on to assume that the epigenetic profiles in the male and female germline will be similar (note – this is a real limitation in the field and does not represent a failing by the authors). Given this limitation, it is possible that the CO frequency/distribution correlations made on the female side do not represent the true biology of the female meiocyte which undermines some of the central conclusions of the manuscript.

Minor Concerns

1. There are a lot of grammar and English usage problems throughout the manuscript, but I assume these can be fixed in production.
2. The authors failed to look for sex-based differential usage of sequence motifs associated with CO hotspots that have been identified by the Henderson and Levy labs.
3. The use of exaggerative terminology is really annoying. What is “ultra-resolution” CO mapping? How is that quantitatively different than “high-resolution”? Later in the manuscript the authors aren’t even satisfied with “ultra” and move on to describing their technique as “ultimate resolution”. These terms have no scientific value and should be expunged throughout the manuscript.
4. Line 48 – it should be noted that the 200 – 500 DSB estimate for maize is based on indirect measures (protein foci counting) rather than the direct measures used in other organisms such as yeast and mice.
5. Lines 51 – 53: this is a bit strong since there are other organisms, like *S. cerevisiae* which have interference but also have abundant COs.
6. Line 57: while it is true class I COs experience interference when class II COs are nearby, it isn’t clear that class II COs EXERT interference ON class I COs (which implies a directionality).
7. Lines 70-72: *Drosophila* CENs replicate early, but are still CO silent (PMID: 11285277), and maize CENs replicate in mid S-phase (PMID 28842533), so it isn’t clear the argument presented here is a good one.
8. Lines 120-121: the COs/meiosis #s in males and females appear to be reversed.
9. Line 151: were these 6 COs actually WITHIN the rDNA array or could they have been in the

DNA immediately flanking the array?

10. Lines 348-350: reference 48 clearly claims that CEN COs increase in met1 so this text needs to be fixed.

Response to reviewers.

We would like to thank the reviewers for their helpful comments and suggestions. Please see our detailed responses bellow.

Reviewer #1:

1. *My only concern is that the paper is written to emphasize the few areas in which the authors found male/female differences. From my point of view the big story is how similar male and female are, especially as compared to other species that show very big male/female differences in overall recombination count, gross locations of recombination, etc. I suspect that the authors felt like differences are higher-impact than similarities, but the analyses that show differences between males and females seem to me to be the weakest ones, with most statistical tests separately comparing males and females to some "norm" rather than directly comparing them to each other. I would not insist that the authors revise the manuscript based on these comments, but I would encourage them to think about whether the bigger story here is the differences or the similarities.*

We agree that this is a very good point. Following reviewer's suggestion, in the revised manuscript we strived to equally emphasize similarities and differences between male and female CO landscapes, which we believe reflects our data better.

2. *The claim that GO analysis shows "different classes of genes" near male and female crossovers seems particularly overblown to me.*

This comment is similar to that of Reviewer 2. Please see the response bellow.

Reviewer #2:

Is there any functional significance in the fact that crossover hotspots in males are enriched in association with a different group of genes than hotspots in females?

Much has been written about the role of recombination in creating genetic diversity that facilitates evolution of certain rapidly diversifying genes. The best example of this phenomenon in plants is the fact that several recombination hotspots in Arabidopsis are associated with disease resistance genes. This observation suggests that recombination patterns in the Arabidopsis genome are specifically arranged to facilitate rapid evolution of these genes within the frame of host-pathogen coevolution ("arms race"). Interestingly, observations from other plants, including a recent study in potato as well as our data, indicated that other groups of genes were significantly enriched at recombination sites. Thus, it is likely that which genes become sites of recombination hotspots is not universal among species but a response to specific selective pressures. It may be tempting to speculate that because maize and potato are crops, their domestication and breeding may affect which regions of their genome experience pressure to diversify, although data to test this hypothesis are lacking as of yet. In view these data, we believe that it is interesting to report statistically significant differences in gene classes enriched at CO sites in male and female, even though the functional relevance of these differences is unclear at the moment. We revised the manuscript to give more context on why we report these data and remove the lengthy discussion of this issue from the Discussion section.

Reviewer #3:

1. *The study is entirely descriptive. This kind of genomic profiling is an appropriate starting place for querying sex differences in CO control, but given the rich genetic resources available in maize follow-on experimentation to test the emerging hypotheses is within the expectations of the maize genetics community.*

Our ability to alter recombination patterns, to be able to provide the follow-up that the reviewer suggests, is unfortunately limited in maize. The two main ways of altering genome-wide recombination patterns in Arabidopsis, which is employing mutants in anti-recombination genes and modifying DNA methylation patterns, are generally not available in maize. Characterized mutants in anti-recombination gene homologs in maize do not exist and the vast majority of mutants in DNA and chromatin modification genes are embryo lethal in maize. However, we hope that our addition of data from the B73 x CML228 maize hybrid (please see below) will go some way towards examining some of the hypotheses generated by the detailed examination of the B73 x Mo17 hybrid.

2. *All of the conclusions are based on crosses between B73 and Mo17. It is entirely possible that the correlations that the authors describe are idiosyncratic to that cross and do not represent maize broadly, much less other plant species. This significantly reduces the importance of the findings for the broad genetics community.*

We agree with the reviewer that analyses of additional populations and species would be helpful. Unfortunately, such analyses are very challenging in non-model species, particularly those with large genomes. We believe that a solid assembled genome scaffold is critical for examining CO patterns to avoid the caveats of mistaking structural variation for CO events. To satisfy reviewer's comment, we analysed female meiosis from another maize hybrid B73 x CML228 and included the results in the revised manuscript.

3. *The study surpasses many similar prior studies in other plants that relied on comparing somatic epigenetic profiles to meiotic CO frequencies/distributions. In this study the authors made the effort to provide a more direct comparison to epigenetic profiles from male meiocytes – this was a real strength for the paper. Even so, the authors go on to assume that the epigenetic profiles in the male and female germline will be similar (note – this is a real limitation in the field and does not represent a failing by the authors). Given this limitation, it is possible that the CO frequency/distribution correlations made on the female side do not represent the true biology of the female meiocyte which undermines some of the central conclusions of the manuscript.*

We agree with the reviewer and do look forward to when such studies will be feasible in plants.

4. *There are a lot of grammar and English usage problems throughout the manuscript, but I assume these can be fixed in production.*

We have carefully edited the manuscript to correct grammar and improve English usage.

5. *The authors failed to look for sex-based differential usage of sequence motifs associated with CO hotspots that have been identified by the Henderson and Levy labs.*

In the revised manuscript version, we included analyses of DNA sequence motifs associated with COs. Briefly, we found three types of sequence motifs, similar to these that have been previously identified in maize and Arabidopsis. However, we did not find differences in motif usage between male and female.

6. *The use of exaggerative terminology is really annoying. What is “ultra-resolution” CO mapping? How is that quantitatively different than “high-resolution”? Later in the manuscript the authors aren’t even satisfied with “ultra” and move on to describing their technique as “ultimate resolution”. These terms have no scientific value and should be expunged throughout the manuscript.*

Our initial idea of using terms other than “high-resolution” was to distinguish our dataset from several other studies in which the term “high-resolution” was overused to describe mapping resolution that we would not consider “high”. Nevertheless, we agree that by using “ultra-resolution” or alike, we would commit the same error. Consequently, we used the widely-accepted term “high-resolution” in the revised manuscript.

7. *Line 48 – it should be noted that the 200 – 500 DSB estimate for maize is based on indirect measures (protein foci counting) rather than the direct measures used in other organisms such as yeast and mice.*

We included a disclaimer explaining that the DSB estimates are based on RAD51 foci counts.

8. *Lines 51 – 53: this is a bit strong since there are other organisms, like *S. cerevisiae* which have interference but also have abundant COs.*

Following reviewer’s suggestion, we replaced the word “limited” in the sentence in question with “discouraged”.

9. *Line 57: while it is true class I COs experience interference when class II COs are nearby, it isn’t clear that class II COs EXERT interference ON class I COs (which implies a directionality).*

We revised the sentence to avoid this incorrect implication.

10. *Lines 70-72: *Drosophila* CENs replicate early, but are still CO silent (PMID: 11285277), and maize CENs replicate in mid S-phase (PMID 28842533), so it isn’t clear the argument presented here is a good one.*

We have revised our statement to make it more precise that we are referring to pericentromeric regions. The broad regions around the centromere, which were the focus of the Higgins et al. analyses, lack COs in maize and barley, and have been shown to replicate late by multiple studies.

11. *Lines 120-121: the COs/meiosis #s in males and females appear to be reversed.*

Indeed. We corrected the sentence.

12. *Line 151: were these 6 COs actually WITHIN the rDNA array or could they have been in the DNA immediately flanking the array?*

The COs are in the rDNA region but not within the rDNA array. We rephrased the statement in the revised manuscript to make this point clear.

13. *Lines 348-350: reference 48 clearly claims that CEN COs increase in *met1* so this text needs to be fixed.*

We revised the statement to correct it.

Reviewers' Comments:

Reviewer #1 (Remarks to the Author):

The revised manuscript does a good job of toning down some of the language that initially felt exaggerated. I also like the way that male/female differences and similarities are discussed in the revised manuscript.

Reviewer #2 (Remarks to the Author):

In this revised manuscript Dr Pawlowski and colleagues have taken into account the comments from the first round review. They have dealt with the specific comments as far as can reasonably be expected. Overall, this is a a very nice, careful piece of work that makes a very useful contribution to our understanding of the factors that influence the distribution of genetic crossovers in male and female reproductive tissues of maize, which is one of the most important crops grown around the world.

Based on these revisions I believe it now merits publication.

Referee #3 is unable to review the revised manuscript. We asked Referee #2, who also works on plant cross over/meiosis, to comment whether (s)he thinks Referee #3's concerns have been successfully addressed. His/her suggestions is attached with this email.

Reviewer #2 – comments of authors response to Reviewer #3

Reviewer #3:

1. The study is entirely descriptive. This kind of genomic profiling is an appropriate starting place for querying sex differences in CO control, but given the rich genetic resources available in maize follow-on experimentation to test the emerging hypotheses is within the expectations of the maize genetics community.

Our ability to alter recombination patterns, to be able to provide the follow-up that the reviewer suggests, is unfortunately limited in maize. The two main ways of altering genome-wide recombination patterns in Arabidopsis, which is employing mutants in anti-recombination genes and modifying DNA methylation patterns, are generally not available in maize. Characterized mutants in anti-recombination gene homologs in maize do not exist and the vast majority of mutants in DNA and chromatin modification genes are embryo lethal in maize. However, we hope that our addition of data from the B73 x CML228 maize hybrid (please see below) will go some way towards examining some of the hypotheses generated by the detailed examination of the B73 x Mo17 hybrid.

The reviewer's comment was similar to one of mine that the study does not address underlying mechanisms. That said, I think the work makes an important contribution as it stands. Sorting out any underlying mechanism(s) will be interesting as reviewer 3 correctly states but will be a substantial project in its' own right. Hence I don't think what the reviewers asks is reasonable with the context of this piece of work and that the inclusion of the additional data by the authors is a sufficient and reasonable response.

2. All of the conclusions are based on crosses between B73 and Mo17. It is entirely possible that the correlations that the authors describe are idiosyncratic to that cross and do not represent maize broadly, much less other plant species. This significantly reduces the importance of the findings for the broad genetics community.

We agree with the reviewer that analyses of additional populations and species would be helpful. Unfortunately, such analyses are very challenging in non-model species, particularly those with large genomes. We believe that a solid assembled genome scaffold is critical for examining CO patterns to avoid the caveats of mistaking structural variation for CO events. To satisfy reviewer's comment, we analysed female meiosis from another maize hybrid B73 x CML228 and included the results in the revised manuscript.

I think the authors response is satisfactory they have included additional data which addresses the reviewers comments. He authors are correct these analyses are non-trivial for the these large genome species, hence this work is an important step.

3. The study surpasses many similar prior studies in other plants that relied on comparing somatic epigenetic profiles to meiotic CO frequencies/distributions. In this study the authors made the effort to provide a more direct comparison to epigenetic profiles from male meiocytes –this was a real strength for the paper. Even so, the authors go on to assume that the epigenetic profiles in the male and female germline will be similar (note – this is a real limitation in the field and does not represent a failing by the authors). Given this limitation, it is possible that the CO frequency/distribution correlations made on the female side do not represent the true biology of the female meiocyte which undermines some of the central conclusions of the manuscript.

We agree with the reviewer and do look forward to when such studies will be feasible in plants.

This is not really a direct criticism by the reviewer. He/She simply comment on the limitations that currently prevail in this context

4. There are a lot of grammar and English usage problems throughout the manuscript, but I assume these can be fixed in production.

We have carefully edited the manuscript to correct grammar and improve English usage. *The authors have satisfactorily addressed this point.*

5. The authors failed to look for sex-based differential usage of sequence motifs associated with CO hotspots that have been identified by the Henderson and Levy labs.

In the revised manuscript version, we included analyses of DNA sequence motifs associated with COs. Briefly, we found three types of sequence motifs, similar to these that have been previously identified in maize and Arabidopsis. However, we did not find differences in motif usage between male and female.

The authors have satisfactorily addressed this point. Although the outcome is negative, it is an important piece of knowledge for the community.

6. The use of exaggerative terminology is really annoying. What is “ultra-resolution” CO mapping? How is that quantitatively different than “high-resolution”? Later in the manuscript the authors aren’t even satisfied with “ultra” and move on to describing their technique as “ultimate resolution”. These terms have no scientific value and should be expunged throughout the manuscript.

Our initial idea of using terms other than “high-resolution” was to distinguish our dataset from several other studies in which the term “high-resolution” was overused to describe mapping resolution that we would not consider “high”. Nevertheless, we agree that by using “ultraresolution” or alike, we would commit the same error. Consequently, we used the widely accepted term “high-resolution” in the revised manuscript.

The authors have satisfactorily addressed this point.

7. Line 48 – it should be noted that the 200 – 500 DSB estimate for maize is based on indirect measures (protein foci counting) rather than the direct measures used in other organisms such as yeast and mice.

We included a disclaimer explaining that the DSB estimates are based on RAD51 foci counts.

The authors have satisfactorily addressed this point. The authors are correct in pointing out that there is good correlation between RAD51 foci numbers (and foci of other proteins recruited to DNA double-strand break sites and DSB numbers assessed by direct methods.

8. Lines 51 – 53: this is a bit strong since there are other organisms, like *S. cerevisiae* which have interference but also have abundant COs.

Following reviewer’s suggestion, we replaced the word “limited” in the sentence in question with “discouraged”.

*The authors have satisfactorily addressed this point. Although, its worth pointing out that *S. cerevisiae* is a little unusual in that it has high levels of COs - most species are similar to maize where interference tends to result in few COs.*

9. Line 57: while it is true class I COs experience interference when class II COs are nearby, it isn't clear that class II COs EXERT interference ON class I COs (which implies a directionality). We revised the sentence to avoid this incorrect implication.

The authors have satisfactorily addressed this point.

10. Lines 70-72: Drosophila CENs replicate early, but are still CO silent (PMID: 11285277), and maize CENs replicate in mid S-phase (PMID 28842533), so it isn't clear the argument presented here is a good one.

We have revised our statement to make it more precise that we are referring to pericentromeric regions. The broad regions around the centromere, which were the focus of the Higgins et al. analyses, lack COs in maize and barley, and have been shown to replicate late by multiple studies.

The authors have satisfactorily addressed this point.

11. Lines 120-121: the COs/meiosis #s in males and females appear to be reversed.

Indeed.

We corrected the sentence.

The authors have satisfactorily addressed this point.

12. Line 151: were these 6 COs actually WITHIN the rDNA array or could they have been in the DNA immediately flanking the array?

The COs are in the rDNA region but not within the rDNA array. We rephrased the statement in the revised manuscript to make this point clear.

The authors have satisfactorily addressed this point.

13. Lines 348-350: reference 48 clearly claims that CEN COs increase in met1 so this text needs to be fixed.

We revised the statement to correct it.

The authors have satisfactorily addressed this point.